# Monoclonal Antibodies Targeting Alarmins: A New Perspective for Biological Therapies of Severe Asthma

**DOI:** 10.3390/biomedicines9091108

**Published:** 2021-08-29

**Authors:** Corrado Pelaia, Giulia Pelaia, Federico Longhini, Claudia Crimi, Cecilia Calabrese, Luca Gallelli, Angela Sciacqua, Alessandro Vatrella

**Affiliations:** 1Department of Health Sciences, University “Magna Graecia” of Catanzaro, Viale Europa-Località Germaneto, 88100 Catanzaro, Italy; gallelli@unicz.it; 2Department of Medical and Surgical Sciences, University “Magna Graecia” of Catanzaro, 88100 Catanzaro, Italy; giulia.pelaia@gmail.com (G.P.); longhini.federico@gmail.com (F.L.); sciacqua@unicz.it (A.S.); 3Department of Clinical and Experimental Medicine, University of Catania, 95131 Catania, Italy; dott.claudiacrimi@gmail.com; 4Department of Translational Medical Sciences, University of Campania “Luigi Vanvitelli”, 80131 Naples, Italy; cecilia.calabrese@unicampania.it; 5Department of Medicine, Surgery, and Dentistry, University of Salerno, 84084 Salerno, Italy; avatrella@unisa.it

**Keywords:** asthma, TSLP, IL-33, IL-25, tezepelumab, anti-alarmins

## Abstract

Alarmins are innate cytokines, including thymic stromal lymphopoietin (TSLP), interleukin-33 (IL-33), and interleukin-25 (IL-25), which are mainly produced by airway epithelium and exert a prominent role in asthma pathobiology. In particular, several environmental factors such as allergens, cigarette smoking, airborne pollutants, and infectious agents trigger the release of alarmins, which in turn act as upstream activators of pro-inflammatory pathways underlying type 2 (T2-high) asthma. Indeed, alarmins directly activate group 2 innate lymphoid cells (ILC2), eosinophils, basophils, and mast cells and also stimulate dendritic cells to drive the commitment of naïve T helper (Th) cells towards the Th2 immunophenotype. Therefore, TSLP, IL-33, and IL-25 represent suitable targets for add-on therapies of severe asthma. Within this context, the fully human anti-TSLP monoclonal antibody tezepelumab has been evaluated in very promising randomized clinical trials. Tezepelumab and other anti-alarmins are thus likely to become, in the near future, valuable therapeutic options for the biological treatment of uncontrolled severe asthma.

## 1. Introduction

Asthma is a chronic inflammatory disease of the respiratory tract, usually characterized by variable airflow limitation and affecting more than 300 million people worldwide [1,2]. Asthma phenotypes are heterogeneous and include eosinophilic patterns, underpinned by either allergic or non-allergic mechanisms, as well as neutrophilic and paucigranulocytic traits [3,4,5]. Eosinophilic asthma is driven by complex cellular and molecular pathways which shape the so-called type 2 (T2-high) airway inflammation, a term based on the key pathogenic roles played by T helper 2 (Th2) lymphocytes and group 2 innate lymphoid cells (ILC2), releasing the interleukins 4 (IL-4), 5 (IL-5), 9 (IL-9), and 13 (IL-13) [6,7,8]. Within this pathobiological context, the bronchial epithelium exerts pivotal functions by producing three innate cytokines named alarmins, which include thymic stromal lymphopoietin (TSLP), interleukin-33 (IL-33), and interleukin-25 (IL-25) [9,10,11]. The secretion of these cytokines occurs when airway epithelial cells undergo injuries caused by several environmental triggers such as allergens, respiratory viruses, bacteria, cigarette smoking, and airborne pollutants (Figure 1) [12]. Therefore, TSLP, IL-33, and IL-25 provide alarm signals which solicit the immune system to react to bronchial epithelial damage [9]. As a consequence, these alarmins behave as upstream activators of both innate and adaptive immune responses involved in type 2 asthma (Figure 1). Indeed, alarmins are able to induce conventional dendritic cells to drive the differentiation and expansion of Th2 cells, as well as to directly promote ILC2 activation and proliferation [13]. Moreover, by stimulating the commitment of naïve Th lymphocytes towards the Th17 cellular immunophenotype, TSLP can also contribute to the pathogenesis of T2-low neutrophilic asthma [14].

Because of their relevant pathogenic functions, alarmins represent suitable molecular targets for emerging biological therapies of asthma [15,16,17,18,19,20]. In particular, neutralization of alarmins via specific antibodies potentially provides an effective therapeutic strategy capable of interfering at a very upstream level with the complex immunopathologic cascades leading to type 2 asthma. In fact, this approach can impact more broadly on asthma pathobiology with respect to current downstream therapies targeting circumscribed pathways mediated by single cytokines (IL-4, IL-5, and IL-13) or receptors (IL-4 receptor and IL-5 receptor) [21]. Among anti-alarmins, the most studied anti-TSLP monoclonal antibody in clinical trials is tezepelumab [20,22,23]; some anti-IL-33 antibodies are under investigation, whereas no anti-IL-25 drug has entered clinical evaluation so far [18]. In this regard, it is noteworthy that anti-alarmins could induce effective therapeutic benefits for patients with uncontrolled T2-high asthma by lowering disease exacerbations, as well as by improving symptom control, lung function, quality of life, and oral corticosteroid (OCS) dependence [18,20]. Furthermore, tezepelumab should also be evaluated in patients with difficult-to-treat T2-low asthma.

Therefore, the present review article focuses on the following topics: (i) elucidation of alarmin roles in asthma pathobiology; and (ii) discussion of the potential therapeutic effects of anti-alarmins in severe asthma. In order to write this narrative review, a literature search was carried out using the PubMed database, and as search words, we chose ‘asthma’, ‘alarmins’, ‘TSLP’, ‘IL-33′, ‘IL-25′, ‘anti-alarmins’, and ‘tezepelumab’.

## 2. Pathogenic Roles of Alarmins in Asthma

### 2.1. TSLP

First discovered in thymic stromal cells, TSLP belongs to the interleukin-2 (IL-2) family, and its structure is related to interleukin-7 (IL-7) [24]. Two TSLP isoforms exist, characterized by longer (159 amino acids) and shorter sequences (60 amino acids) [25]. The short variant is constitutively expressed and exerts homeostatic functions, whereas the synthesis of long TSLP can be stimulated by inflammatory triggers and is up-regulated in asthmatic patients [26]. The positive charges of the long TSLP surface drive its interaction with the negative charges of the TSLP receptor (TSLPR) [27,28]. As a consequence, the TSLP/TSLPR complex incorporates the α subunit of IL-7 receptor (IL-7Rα), and the result of this sequential process is the assembly of the ternary molecular complex TSLP/TSLPR/IL-7Rα. The latter activates an intricate network of signaling pathways, including Janus kinases 1 and 2 (JAK1/2), signal transducers and activators of transcription 3 and 5 (STAT3/5), and also mitogen-activated protein kinases (MAPK), phosphoinositide 3 kinase (PI3K), and nuclear factor κB (NF-κB), [26,27,28].

In asthmatic patients, TSLP is overexpressed in biopsy specimens from the bronchial epithelium and also in induced sputum, exhaled breath condensate, bronchoalveolar lavage fluid (BALF), and serum [29,30,31,32]. TSLP expression within the airways is correlated with the severity of asthma and airflow limitation [29]. Furthermore, interesting studies based on genome-wide association strategies have shown that some single-nucleotide polymorphisms (SNPs) detected within the TSLP gene are coupled with the relevant asthma risk [33,34]. It is also noteworthy that TSLP mRNA levels are elevated in nasal polyp tissues taken from subjects with aspirin-exacerbated respiratory disease (AERD) [35].

Upon its release from injured airway epithelial cells, TSLP acts on several cellular targets that co-express TSLPR and IL-7Rα (Figure 1). In particular, TSLP enhances the survival of ILC2 and promotes the secretion of IL-4, IL-5, IL-9, and IL-13 from these cells [29,36,37,38]. IL-4 drives Th2 cell differentiation and IgE synthesis, IL-5 is responsible for eosinophilic inflammation, IL-9 stimulates mast cell growth, and IL-13 induces goblet cell hyperplasia as well as both hyperresponsiveness and proliferation of airway smooth muscle cells [36,39]. Within the airways, TSLP co-localizes with ILC2, and a direct relationship has been shown between TSLP levels and ILC2 numbers in nasal biopsy samples taken from patients with chronic rhinosinusitis and severe asthma [40,41]. Moreover, TSLP makes airway ILC2 resistant to corticosteroids [42].

With regard to innate immunity and in addition to ILC2, eosinophils and eosinophil progenitors also express TSLPR and IL17-Rα, thus being subject to the biological effects of TSLP, which provides a powerful anti-apoptotic signal for eosinophils thereby prolonging their survival, as well as potentiates the secretion of eosinophilic cytotoxic proteins and pro-inflammatory chemokines [43,44]. All these pro-eosinophilic effects of TSLP are driven by signaling mechanisms mediated by MAPK and NF-κB activation [43,44]. Moreover, 24 h after the aeroallergen challenge, TSLP expression co-localizes with airway eosinophilia in asthmatic patients [45]. TSLP also induces the build-up of eosinophilic extracellular traps, consisting of aggregates including eosinophilic cationic protein (ECP) and mitochondrial DNA, which are involved in innate immune responses inside asthmatic airways [46]. Eosinophil progenitors are further cellular targets of TSLP, which up-regulates the expression levels of the α subunit of the IL-5 receptor (IL-5Rα), thereby stimulating together with IL-5 the maturation of eosinophils [47]. TSLP not only induces eosinophilopoiesis but also promotes the trafficking of eosinophil progenitors by increasing their production of several chemokines, including CXCL8, CCL1, and CCL22 [29,48].

Other cellular elements belonging to the innate immune system, whose functions are affected by TSLP, include basophils and mast cells (Figure 1) [20]. At the level of basophils, TSLP induces cell differentiation, stimulates histamine release, promotes cytokine production, and up-regulates the expression of the basophil activation marker CD203c [29,49]. In allergic asthmatic patients, TSLP enhances basophil surface levels of both IL-17RB receptor of IL-25 and ST2 receptor of IL-33, thereby increasing the responsiveness of basophils to other alarmins [50]. Moreover, mast cell secretion of IL-5 and IL-13 can be elicited by TSLP, which acting together with IL-33 also increments mast cell production of prostaglandin D_2_ (PGD_2_), a multifunctional eicosanoid implicated in the pathobiology of type 2 asthma [35,51,52].

With regard to the pathophysiology of asthma, TSLP plays a very important role in the complex interplay between the innate and adaptive branches of the immune system. In particular, TSLP is significantly involved in the intercellular interactions occurring between dendritic cells and Th2 lymphocytes (Figure 1). Upon release from damaged bronchial epithelium, TSLP acts on TSLPR+ human myeloid dendritic cells by up-regulating the expression of major histocompatibility complex class II (MHC-II) antigens, as well as of co-stimulatory molecules, including CD40 and CD86 [53]. Furthermore, under TSLP-dependent stimulation, dendritic cells increase the secretion of CCL17 and CCL22, two chemokines that bind to CCR4 receptors expressed by Th2 lymphocytes, thus promoting their migration from thoracic lymph nodes towards the bronchial tree [29,54]. A relevant function is exerted by TSLP as a key inducer of Th2 cell differentiation driven by dendritic cells [55]. Indeed, TSLP up-regulates the expression of OX40 ligand (OX40L) on dendritic cells, which respond to this stimulation by promoting the commitment of naïve CD4^+^ T cells towards the Th2 lineage [56]. Following TSLP-dependent activation, OX40L+ dendritic cells also induce the development in lung-draining lymph nodes of T follicular helper cells (Tfh) releasing IL-4 [57], an essential cytokine for Th2 cell maturation and IgE production [13]. In addition to dendritic cells, other cellular populations such as CD11c^+^ monocytes/interstitial macrophages can mediate the stimulatory effects of TSLP on Th2 cell differentiation [58]. However, although T lymphocytes are often indirectly regulated by TSLP through dendritic cells and monocytes/macrophages, this alarmin can also directly affect the functions of CD4^+^ T cells by inducing their polarization towards the Th2 immunophenotype [59]. In allergic asthmatic patients, a further contribution to the expansion of type 2 pro-inflammatory responses is given by TSLP via its inhibitory effect on immune modulation provided by lung T regulatory (Treg) cells [60]. Indeed, TSLP acts on Treg lymphocytes by down-regulating their biosynthesis of interleukin-10 (IL-10), an anti-allergic and anti-inflammatory cytokine responsible for the suppression of Th2 cell-driven immune pathways [60].

In addition to being involved in the pathogenesis of T2-high bronchial inflammation, TSLP seems to be also implicated in type 2-low neutrophilic asthma, which is highly dependent on the pro-inflammatory mechanisms triggered by Th17 cells [61]. In fact, TSLP can stimulate dendritic cells to secrete interleukin-6 (IL-6) and interleukin-23 (IL-23), whose role is critical in the induction of Th17 cell signature [14,62]. Moreover, TSLP can also lead to a dual concomitant amplification of Th2 and Th17 immune profiles, associated with the production of IL-4 and IL-17A, respectively [62].

TSLP not only regulates the bioactivities of immune/inflammatory cells but also affects the functions of bronchial structural cells. Indeed, in asthmatic patients, TSLP is overexpressed in airway epithelial cells, where this alarmin induces the production of IL-13, thus eliciting a proliferative and reparative response, mediated by an IL-13-dependent autocrine loop [63]. Additionally, TSLP promotes the release of IL-6 and IL-8 from airway smooth muscle cells [64,65]. The latter are stimulated by adjacent activated mast cells to produce TSLP itself, interleukin-1β (IL-1β), and tumor necrosis factor-α (TNF-α) [29,66]. Moreover, airway fibroblasts synthesize TSLP and, upon autocrine stimulation mediated by TSLP, enhance their production of collagen, arginase 1, α smooth muscle actin, and transforming growth factor β1 (TGF-β1) [67,68]. Therefore, these findings imply that TSLP is significantly involved in the crosstalk between inflammatory and resident cells, leading to the structural changes which characterize airway remodeling in asthma.

### 2.2. IL-33

IL-33 belongs to the IL-1 family of cytokines and plays a pivotal role in the pathophysiology of type 2 asthma [9,16]. The biological actions of IL-33 are mediated by its binding to a heterodimeric receptor consisting of the ST2 (suppression of tumorigenicity 2) component, coupled to its co-receptor IL-1RAcP (IL-1 receptor accessory protein) [18]. The interaction between IL-33 and its receptor is followed by the recruitment of a signaling module including the adaptor protein MyD88 (myeloid differentiation primary response protein 88), the IL-1 receptor-associated kinase (IRAK), and the TNF receptor-associated factor 6 (TRAF6), which in turn activate downstream kinases and transcription factors such as MAPK, PI3K, and NF-κB [69].

When compared to healthy subjects, the amounts of IL-33 and ST2 resulted in being significantly elevated in serum and bronchial biopsies taken from asthmatic patients, and the expression levels of this alarmin and its receptor are directly correlated with asthma severity [70,71,72]. In particular, high serum levels of the soluble form of ST2 (sST2) are reliable predictors of the occurrence of asthma exacerbations within 3 months [73]. Moreover, in comparison with T2-low asthma, type 2 disease is associated with higher IL-33 serum levels [74]. Additionally, inhaled allergen challenge enhances bronchial expression of both IL-33 and ST2 receptors in allergic asthmatic patients [75,76].

After being released from the injured bronchial epithelium, IL-33 can affect the functions of several immune/inflammatory and structural cells. With regard to the crosstalk between innate and adaptive immune responses, underlying the pathobiology of type 2 asthma, IL-33 promotes the polarization and expansion of Th2 lymphocytes and also stimulates ILC2 production of IL-5 and IL-13 (Figure 1) [77]. IL-33 favors the development of allergic asthma by impairing the immunologic tolerance to inhaled allergens operated by Treg cells, whose immunomodulatory action is suppressed by this alarmin [78,79]. Moreover, IL-33 activates eosinophils, prolongs eosinophil survival, and enhances the expression of eosinophil adhesion molecules [77,80]. IL-33 induces the growth, maturation, and survival of mast cells and also elicits the production of IL-4 and IL-13 from basophils [81,82]. In addition to recruiting mast cells, basophils, and eosinophils into the airways, IL-33 also promotes the release of fibrogenic factors from these cells [83]. In particular, within the context of chronic allergic/eosinophilic inflammation, this alarmin induces the production of amphiregulin, which in turn promotes the synthesis of osteopontin and the consequent fibrotic response of the airways [84]. IL-33 further contributes to the pathogenesis of airway remodeling in asthma by activating the epithelial–mesenchymal trophic unit, as well as by stimulating fibroblasts to produce extracellular matrix proteins, such as collagen type 1 and fibronectin [79,85].

### 2.3. IL-25

With respect to healthy subjects, asthmatic patients express higher serum levels of IL-25, also known as IL-17E, which belongs to the IL-17 cytokine family and exerts its biological effects by interacting with a dimeric complex consisting of the two receptor subunits IL-17RA and IL-17RB [8,86,87]. Binding of IL-25 to IL-17RA/IL-17RB induces the recruitment of the adaptor protein Act1 (activator of NF-κB) and the subsequent activation of a signal transduction network, including TRAF4/TRAF6 and NF-κB, as well as the JAK1/2-STAT3 module [87,88].

Although Th2 lymphocytes, mast cells, basophils, eosinophils, alveolar macrophages, and fibroblasts can produce IL-25, the main cellular sources of this cytokine are airway epithelial cells [10,87]. IL-25 is stored within the cytoplasm of bronchial epithelial cells, which release it upon exposure to aeroallergens with intrinsic proteolytic bioactivity such as house dust mite [89]. In comparison to non-allergic asthmatic patients, subjects with atopic asthma have higher sputum levels of IL-25 [16]. Indeed, IL-25 exerts a relevant pathogenic role in allergic asthma and virus-induced exacerbations of type 2 airway inflammation [90]. In particular, IL-25 is overexpressed in the airway epithelium of patients with type 2 asthma, whose plasma IL-25 levels correlate with the therapeutic response to inhaled corticosteroids (ICS) [91]. The stimulatory action exerted by IL-25 on type 2 inflammation is associated with the inhibitory effect of this alarmin on differentiation and activation of Th1 and Th17 lymphocytes [92].

IL-25 stimulates ILC2 and Th2 cells to secrete IL-4, IL-5, IL-9, and IL-13 [10]. Other cellular targets activated by IL-25 include dendritic cells, basophils, and eosinophils (Figure 1). With regard to these latter cells, IL-25 inhibits the apoptosis of eosinophils, thus prolonging their survival [93]. In addition to activating immune/inflammatory cells, IL-25 also affects the functions of airway structural cells. In particular, IL-25 triggers an autocrine mechanism leading to its increased biosynthesis from bronchial epithelial cells, induces airway smooth muscle cell contraction, and contributes to remodeling changes by stimulating fibroblast proliferation and the production of extracellular matrix proteins [86,87].

In addition to asthma, IL-25 plays an important pathogenic role also in nasal polyposis. Indeed, IL-25 is highly expressed in nasal polyp tissues, where this alarmin amplifies the activation of Th2 and ILC2 cells, thereby worsening both endoscopic nasal polyp score and computed tomography score [94].

## 3. Anti-Alarmins

### 3.1. Anti-TSLP

Among the anti-TSLP biologics, the most extensively studied drug is tezepelumab, a fully human monoclonal IgG2λ antibody that interacts with TSLP at the level of its binding site for TSLPR, thus preventing the access of TSLP to its receptor [95]. When tezepelumab was firstly evaluated in comparison to placebo in subjects with mild allergic asthma, who received three monthly intravenous injections (700 mg), this antibody inhibited allergen-induced bronchoconstriction on days 42 and 84 [96]. In addition, tezepelumab prevented the airway response to methacholine, reduced the levels of fractional exhaled nitric oxide (FeNO), and lowered eosinophil counts in both blood and sputum, but it did not modify serum concentrations of IgE [96]. No serious adverse reactions occurred during this trial [96].

Later on, the phase 2b randomized, placebo-controlled, multicenter PATHWAY study was performed from December 2013 to March 2017 at 108 sites widely spread throughout 12 countries [97]. The authors of this trial recruited patients with an age ranging from 18 to 75 years, who were current non-smokers complaining of uncontrolled asthma in spite of a notable therapy, including medium-to-high doses of inhaled corticosteroids (ICS), consisting of 250–500 μg or more than 500 μg/day of dry powder fluticasone propionate or equivalents, associated with a long-acting β_2_-adrenergic agonist (LABA). Asthma was defined as uncontrolled when, with regard to the screening phase, patients reported an ACQ-6 (6-item Asthma Control Questionnaire) score of at least 1.5. During the 12 months before recruitment, enrolled participants had either at least two asthma exacerbations or at least one severe asthma exacerbation requiring hospitalization. Lung function was characterized by reversible airflow limitation, documented by pre-bronchodilator FEV_1_ measurements ranging from 40% to 80% of predicted values, which after inhalation of salbutamol (400 μg) improved by at least 12% and 200 mL with regard to baseline levels. In particular, each of 584 admitted subjects was randomly assigned to one of four study clusters comprising a placebo arm including 148 patients, as well as three groups of people treated every 4 weeks with subcutaneous administrations of tezepelumab, consisting of low (70 mg; 145 subjects), medium (210 mg; 145 participants), and high dosages (280 mg; 146 patients).

The main aim of the PATHWAY trial was to verify at week 52 the impact of tezepelumab on the annualized asthma exacerbation rate (AAER). When compared to placebo, tezepelumab significantly (*p* < 0.001) lowered AAER by 61%, 71%, and 66% at the dosages of 70, 210, and 280 mg, respectively, and also prolonged the time to first exacerbation [97]. These very important effects of tezepelumab on AAER occurred regardless of baseline blood eosinophil numbers [97]. In addition, a post hoc analysis of this study demonstrated that tezepelumab, when used at the dose of 210 mg, reduced AAER in patients with severe asthma independently of the eventual concomitance with nasal polyposis [98]. A further post hoc analysis also showed that tezepelumab decreased asthma exacerbations during all four seasons of the year [99]. With regard to secondary outcomes, at the 52nd week of treatment, tezepelumab significantly improved the ACQ-6 score in the three different dosage subgroups. Tezepelumab also increased pre-bronchodilator FEV_1_ by 150 mL, 110 mL, and 120 mL in the high-dose, medium-dose, and low-dose subgroups, respectively [97]. With regard to the biomarkers of type 2 asthma, tezepelumab significantly reduced serum IgE concentrations, blood eosinophil counts, and FeNO levels, as well as blood levels of periostin, IL-5, IL-13, and thymus and activation-regulated chemokine (TARC) [22,91].

Tezepelumab was characterized by a good safety and tolerability profile; indeed, in the four trial subgroups, there were similar numbers of adverse reactions, which mainly included bronchitis, nasopharyngitis, and headache [97]. In particular, similar percentages of patients experienced at least one adverse event across the placebo arm (62.2%) and the three subgroups treated with high (61.6%), medium (64.8%) and low (66.2%) doses of tezepelumab. The numbers of skin reactions at injection sites were also similar among the four study groups. No anaphylactic reactions occurred. Anti-drug antibodies were found in 8.8% of subjects assigned to the placebo arm, as well as in 2.1%, 4.9%, and 0.7% of recipients of high, medium and low tezepelumab doses, respectively. No neutralizing antibodies were detected.

The program of clinical development of tezepelumab includes several other trials named NAVIGATOR, SOURCE, DESTINATION, and CASCADE [22].

The phase 3 multicenter, randomized, placebo-controlled, and double-blind NAVIGATOR trial was carried out in 1061 patients with severe asthma, aged from 12 to 80 years, who experienced recurrent disease exacerbations [100]. In total, 532 subjects were randomly assigned to receive a placebo, and 529 participants were treated with subcutaneous injections of 210 mg of tezepelumab, performed every 4 weeks for 52 weeks. The study population included patients with both more and less than 300/μL blood eosinophil counts. With regard to the primary study outcome, tezepelumab significantly reduced AAER, regardless of blood eosinophil numbers [100]. Secondary endpoints included lung function, symptom control, and health-related quality of life. When compared to placebo, tezepelumab significantly increased pre-bronchodilator FEV_1_, as well as improved the scores of ACQ-6, ASD (Asthma Symptom Diary), and AQLQ (Asthma Quality of Life Questionnaire) [100]. Moreover, tezepelumab decreased serum IgE concentrations, blood eosinophil counts, and FeNO levels. The most frequent adverse events included nasopharyngitis, upper respiratory tract infections, and headache, which, however, did not differ between placebo and tezepelumab subgroups. Reactions at the level of injection sites were observed in 2.6% of subjects assigned to the placebo arm and in 3.6% of patients treated with tezepelumab [100]. Anti-drug antibodies were detected in 8.3% of participants randomized to receive placebo and in 4.9% of patients undergoing treatment with tezepelumab. Neutralizing antibodies were found in two subjects, including one patient for each subgroup [100].

SOURCE is a further phase 3 multicentre, 48-week, double-blind, randomized, and placebo-controlled study, recruiting 150 patients with severe asthma, undergoing treatment with medium-to-high doses of ICS/LABA associations, integrated by an adjunctive chronic OCS therapy [101]. The primary endpoint of this trial is to verify the eventual OCS-sparing effect of tezepelumab, administered subcutaneously at the dose of 210 mg every 4 weeks [101]. This study’s goal is very relevant because of the chronic OCS utilization by many severe asthmatic patients, who can frequently experience the well-known systemic adverse effects induced by these drugs, comprising adrenal insufficiency, diabetes, hypertension, infections, gastrointestinal disorders, psychiatric disturbances, osteoporosis, bone fracture, cataract, glaucoma, as well as decreased growth in children and adolescents [102].

Outside of the phase 3 PATHFINDER program, finalized to the clinical development of tezepelumab in severe asthma and including both SOURCE and NAVIGATOR trials, DESTINATION is another phase 3, 1-year study, enrolling 960 patients who have already participated in one of the above two trials [103]. Hence, DESTINATION is a long-term extension trial whose primary endpoint is the evaluation of the safety profile of tezepelumab in severe asthmatic patients, comprehensively assessed for two years [103]. A secondary outcome of DESTINATION regards the evaluation of the prolonged impact of tezepelumab on AAER.

The phase 2 placebo-controlled, double-blind, and randomized trial CASCADE recruited patients with inadequately controlled, moderate-to-severe asthma, treated with subcutaneous injections of tezepelumab (210 mg) every 4 weeks for 28 weeks [104]. Enrolment was independent of baseline eosinophilic inflammatory status. In particular, CASCADE main goal was the assessment of the possible anti-inflammatory actions of tezepelumab. In this regard, bronchoscopic biopsy samples were evaluated with the aim of exploring the eventual changes in the inflammatory cellular infiltration of bronchial walls, detectable with respect to baseline at the 28th week of add-on therapy with tezepelumab [104]. Moreover, the potential effects of this drug on airway remodeling were investigated by measuring the thickness of the reticular basement membrane before and after treatment with tezepelumab. A total of 48 patients in the tezepelumab group and 51 in the placebo arm completed this study, which showed that tezepelumab was able to significantly decrease the eosinophilic submucosal infiltration of airways without affecting other immune/inflammatory cell populations such as neutrophils, mast cells, and T lymphocytes [105]. Treatment with tezepelumab did not elicit any change in reticular basement membrane thickness, whereas it reduced bronchial hyperresponsiveness to mannitol [105], an indirect bronhoconstrictive stimulus that degranulates mast cells.

In addition to tezepelumab, another anti-TSLP biologic drug under clinical investigation is CSJ117, a fully human IgG1/λ antibody fragment developed for administration through the inhalational route, whose therapeutic effects are not yet known [11,26,106].

### 3.2. Anti-IL-33

Several anti-IL-33 monoclonal antibodies are under clinical evaluation as potential biological therapies for asthma [18]. REGN3500 has been tested in both phase 1 and phase 2a trials. When used alone, REGN3500 improved asthma control but was not more effective than dupilumab [16]. Moreover, when used in combination with dupilumab, REGN3500 did not further improve the therapeutic effects induced by dupilumab alone [16]. An eventual change in blood eosinophil count represents the main outcome evaluated by a phase 2a study investigating the effects of etokimab (ANB020) in patients with severe eosinophilic asthma [16]. GSK3772847 is under ongoing clinical evaluation in subjects with moderate-to-severe asthma in order to verify in a phase 2a trial the effects of this antibody on asthma control [16]. This drug is also being tested in a phase 2b study enrolling asthmatic patients with allergic fungal disease, aimed to check the eventual effects on FeNO levels and blood eosinophils [16]. AMG282 is currently under investigation in a phase 2b study, whose primary endpoint is the impact of drug treatment on AAER [16]. At present ongoing trials also focus on astegolimab, a monoclonal antibody targeting the ST2 receptor of IL-33 [4]. In this regard, the phase 2b randomized, double-blind, dose-ranging, placebo-controlled study ZENYATTA has recently shown that astegolimab was able to significantly decrease AAER in patients with severe asthma [107]. In addition to the use of monoclonal antibodies, other therapeutic strategies are underway with the aim of inhibiting the biological actions of IL-33. In particular, a fusion protein named IL-33 trap has been developed, resulting from the assembly of the extracellular domains of the ST2 receptor with the co-receptor IL-1RAcP [18,108]. Therefore, this molecular construct captures IL-33 and prevents its binding to the ST2/IL-1RAcP receptor complex expressed by target cells. The anti-inflammatory effects of the IL-33 trap have been shown in experimental pre-clinical models of allergic airway inflammation [108].

### 3.3. Anti-IL-25

Pre-clinical experimental models of rhinovirus (RV)-induced exacerbations of allergic asthma have been used to test the eventual therapeutic efficacy of an anti-IL-25 monoclonal antibody (ABM125) [109]. In particular, preliminary results suggest that ABM125 can suppress the type 2 cytokine response elicited by RV infection in bronchial epithelial cells obtained from patients with eosinophilic asthma, and these RV-infected cells respond to ABM125 by increasing their production of anti-viral interferons [109].

## 4. Potential Therapeutic Advantages of Anti-Alarmins with Respect to Currently Available Biologics

In regard to the management of type 2 severe asthma and due to their peculiar mechanisms of action, anti-alarmins could potentially provide relevant advantages with respect to currently available biologics. For instance, IL-5 inhibitors such as mepolizumab and reslizumab, as well as the IL-5 receptor blocker benralizumab, are very effective for treatment of patients with refractory eosinophilic asthma [39], who however remain exposed to the pro-allergic and pro-remodeling effects of IL-4 and IL-13. On the other hand, the dual IL-4/IL-13 receptor antagonist dupilumab may even worsen blood eosinophilia [110,111]. Hence, it would be strategically better to switch off the alarmins-emanated signals, leading to activation of downstream pro-inflammatory pathways mediated by IL-5, IL-4, and IL-13. Moreover, in spite of an add-on therapy with one of the above mentioned monoclonal antibodies, severe asthmatic patients can still experience the need of high dosages of ICS, and they might also be obliged to continue to take OCS [102]. In this regard, it is noteworthy that TSLP is a potent inducer of the corticosteroid resistance of immune-inflammatory cells [42]. It can thus be argued that anti-alarmins could promote the restoration of corticosteroid responsiveness. Finally, the partial therapeutic activity of currently licensed biologics is limited to type 2 asthma, thereby not interfering with the pathogenic mechanisms underlying T2-low disease [17,19]. Therefore, the potential usefulness of anti-alarmins should be also tested in patients with non-type 2 asthma, given the important pathobiologic role played by TSLP in neutrophilic airway inflammation [20]. 

## 5. Conclusions

Alarmins are suitable molecular targets for prospective biological therapies of asthma. Indeed, these innate cytokines act at an upstream pathogenic level, thus triggering complex cascades of downstream pro-inflammatory pathways [112]. Preliminary trials, especially referring to the evaluation of the TSLP-antagonist tezepelumab, have yielded very promising results with regard to the experimental treatment of type 2 asthma. Moreover, tezepelumab should also be investigated for its potential therapeutic effects in T2-low neutrophilic asthma, possibly due to the interferences with the pathophysiologic actions exerted by TSLP- in IL-17-mediated airway inflammation. In addition to TSLP, IL-33 and IL-25 may also represent further emerging targets for novel anti-asthma treatments. In particular, all these new therapies would be especially suitable for patients with severe allergic asthma, not fully responsive to anti-IgE treatment with omalizumab. Indeed, acting at an upstream pathogenic level by disrupting alarmin-mediated interactions between innate and adaptive immunity, anti-alarmins might be more effective than inhibitors of downstream effectors of allergic responses such as IgE. Therefore, in the future, anti-alarmins might be properly positioned as add-on drugs at step 5 of GINA (Global Initiative for Asthma) guidelines for the stepwise treatment of asthma [113], recommending the use of biologics when asthmatic patients are not adequately controlled by standard therapies. Indeed, because of their specific mechanisms of action anti-alarmins should be superior to anti-IgE, as well as to anti IL-5, anti-IL-5 receptor, and anti-IL-4/IL-13 receptor monoclonal antibodies. Nevertheless, head-to-head comparisons between these biologics and anti-alarmins lack in current scientific literature. To our knowledge, in this regard only the anti-IL-33 monoclonal antibody REGN3500 has been comparatively evaluated, without exhibiting any evident superiority to the dual IL-4/IL-13 receptor blocker dupilumab [16]. However, this study appears to be limited and quite inconclusive. Therefore, further and larger investigations are needed to eventually demonstrate the theoretical superiority of anti-alarmins with respect to the currently available biologics, which target specific downstream effectors, but do not interfere with the upstream pathobiologic circuits activated by alarmins.

The main studies aimed to explore the effects of anti-alarmins in asthma treatment are summarized in Table 1.

## Figures and Tables

**Figure 1 biomedicines-09-01108-f001:**
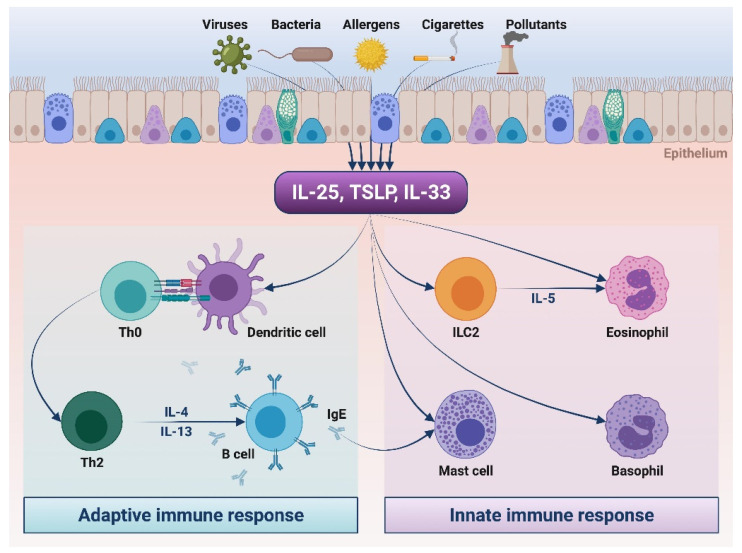
Pathogenic actions of alarmins in type 2 asthma. As a result of injuries caused by several noxious agents, airway epithelial cells release alarmins (TSLP, IL-33, and IL-25). These cytokines induce the activation of the adaptive immune system, thus driving dendritic cell-mediated differentiation of Th2 lymphocytes, leading to B cell-dependent production of immunoglobulins E (IgE). Alarmins also elicit innate immune responses by directly stimulating group 2 innate lymphoid cells (ILC2), eosinophils, basophils, and mast cells. Upon activation induced by alarmins, Th2 and ILC2 cells release high amounts of IL-4, IL-5, and IL-13. This original figure was created by the authors using “BioRender.com” (accessed on 8 August 2021).

**Table 1 biomedicines-09-01108-t001:** Anti-alarmins: summary of the main studies.

Drug—Trial Name	Target	Duration	Main Results or Endpoints
Tezepelumab—PATHWAY [97]	TSLP	52 weeks	Fewer asthma exacerbations, higher ACQ-6 score, FEV_1_ increase.
Tezepelumab—NAVIGATOR [100]	TSLP	52 weeks	Fewer asthma exacerbations, higher ACQ-6 score, FEV_1_ increase.
Tezepelumab—SOURCE [101]	TSLP	48 weeks	OCS-sparing effect.
Tezepelumab—DESTINATION [103]	TSLP	52 weeks	Safety profile, impact on asthma exacerbations.
Tezepelumab—CASCADE [105]	TSLP	28 weeks	Suppression of airway eosinophilic inflammation.
REGN3500 [16]	IL-33	36 weeks	Improvement of asthma control.
AMG282 [16]	IL-33	20 weeks	Impact on asthma exacerbations.
GSK3772847 [16]	IL-33	16 weeks	Improvement of asthma control.
Etokimab [16]	IL-33	9 weeks	Reduction of blood eosinophil count.
Astegolimab—ZENYATTA [107]	ST2	54 weeks	Fewer asthma exacerbations.
ABM125 [109]	IL-25	-	Suppression of type 2 cytokine response induced by rhinovirus.

## Data Availability

Not applicable.

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
