# Peer review of "Monoclonal Antibodies Targeting Alarmins: A New Perspective for Biological Therapies of Severe Asthma"

_biomedicines, 2021, doi:10.3390/biomedicines9091108_

Round 1
Reviewer 1 Report
The revier presented is really interesting, didactic, while also introducing a lot of knowldege baout the topic, that makes it very interesting for newcomers in the matter. Nevertheless I recommend to address some minor changes, and aditions to make the article optimal.
- In lines 173 to 182 it would be nice if you could also mention that TSLP acts autocrinally upon airway epithelial cells themselves, acting in TSLP and IL-13 loop on epithelial cell proliferation and repair (DOI: 10.1016/j.jaci.2010.01.044 or other articles).
- It would be nice if you could add some more information in the role of IL-33 in airway remodeling, mentioning its interaction with amphiregulin (as in DOI: 10.1016/j.immuni.2018.04.023 and others).
- Please briefly mention soluble ST2 receptor and its role in asthma, and as a biomarker in serum (doi: 10.1186/s12931-018-0872-2).
- In line 215 please mention that IL-25 is also known as interleukin-17E (IL-17E) to further clarify.
- Regarding IL-25 please introduce some additional research on IL-25 role in nasal polyps (https://doi.org/10.1111/all.13267), and also mention the direct suppressor effect of IL-25 over Th1/17 immune responses (DOI: 10.1097/MIB.0b013e3182802a76). Also, mention the discovery that airway epithelial cells from asthmatics have differential expression of IL-25, being the expression of IL-25 high or low related to ICS responsiveness (https://doi.org/10.1164/rccm.201403-0505OC).
- Although it is only a poster, it would be nice to mention that there are preclinical trials for anti-IL-25 (ABM125), look for the poster: “ABM125 Anti-IL-25 Antibody Pre-Clinical Development for Viral Asthma Exacerbations Identifies IL-25 Mediated Regulation of Type-2- and Anti-Viral Immunity” American Journal of Respiratory and Critical Care Medicine 2018;197:A7759.
- Figure 1 seems to simplified in my point of view, it would be nice if you could add more information about the actions of the immune cells affected by the alarmins, as the cytokines and other molecules involved.
- I suggest adding a table summarizing the characteristics, endpoints and conclusions from each clinical trial targeting TSLP and IL-33 to make facilitate information flow on this matter.
Author Response
We would like to thank very much the Reviewers for having carefully reviewed our manuscript, thus suggesting those changes which have significantly improved its overall quality. We have prepared a revised version of this paper, taking into account the comments of the 3 reviewers. We also include a point by point response to the criticisms raised by the Referees. The changes have been highlighted in yellow, and deletions have been evidenced in red. Therefore, we hope that our revised manuscript is now suitable for publication in Biomedicines.
REVIEWER 1
In lines 173 to 182 it would be nice if you could also mention that TSLP acts autocrinally upon airway epithelial cells themselves, acting in TSLP and IL-13 loop on epithelial cell proliferation and repair.
This concept has been mentioned in the revised text, and the relative suggested reference has been cited (page 5 of the revised text, lines 178-181).
It would be nice if you could add some more information in the role of IL-33 in airway remodeling, mentioning its interaction with amphiregulin.
This concept has been mentioned in the revised text, and the relative suggested reference has been cited (page 5 of the revised text, lines 218-221).
Please briefly mention soluble ST2 receptor and its role in asthma, and as a biomarker in serum.
This concept has been mentioned in the revised text, and the relative suggested reference has been cited (page 5 of the revised text, lines 202-203).
In line 215 please mention that IL-25 is also known as interleukin-17E (IL-17E) to further clarify.
This change has been made in the revised text (page 6 of the revised text, line 228).
Regarding IL-25 please introduce some additional research on IL-25 role in nasal polyps, and also mention the direct suppressor effect of IL-25 over Th1/17 immune responses.
These concepts have been mentioned in the revised text, and the relative suggested references have been cited (page 6 of the revised text, lines 255-258; page 6 of the revised text, lines 243-245).
Also, mention the discovery that airway epithelial cells from asthmatics have differential expression of IL-25, being the expression of IL-25 high or low related to ICS responsiveness.
This concept has been mentioned in the revised text, and the relative suggested reference has been cited (page 6 of the revised text, lines 241-243).
Although it is only a poster, it would be nice to mention that there are preclinical trials for anti-IL-25 (ABM125), look for the poster.
This concept has been mentioned in the revised text, and the relative suggested reference has been cited (page 9 of the revised text, lines 404-410).
Figure 1 seems to be simplified in my point of view. It would be nice if you could add more information about the actions of the immune cells affected by the alarmins, as the cytokines and other molecules involved.
In the revised manuscript Figure 1 has been integrated by more information, regarding the cytokine secretory pattern of both Th2 and ILC2 cells.
I suggest adding a table summarizing the characteristics, endpoints and conclusions from each clinical trial targeting TSLP and IL-33 to facilitate information flow on this matter.
At the end of the revised text (page 10, lines 435-462), we added a table which summarizes the main studies referring to the evaluation of antibodies targeting alarmins.
Reviewer 2 Report
This is a review article that pursued tezepelumab and other anti-alarmins would probably become a valuable therapeutic options for biological treatment of uncontrolled severe asthma.
This is a scenario review manuscript. I would suggest author to use PRISMA principle (https://www.prisma-statement.org) and re-arrange the manuscript.
On the other hand, let think what is known about biological therapies of severe asthma; did this review give a summary in this issue? with meta-analysis is better, but not required. What new does in this review add? Is it systemic or enough information to persuade peer experts?
Author Response
We would like to thank very much the Reviewers for having carefully reviewed our manuscript, thus suggesting those changes which have significantly improved its overall quality. We have prepared a revised version of this paper, taking into account the comments of the 3 reviewers. We also include a point by point response to the criticisms raised by the Referees. The changes have been highlighted in yellow, and deletions have been evidenced in red. Therefore, we hope that our revised manuscript is now suitable for publication in Biomedicines.
REVIEWER 2
This is a scenario review manuscript. I would suggest author to use PRISMA principle and re-arrange the manuscript.
The PRISMA method applies especially to systematic reviews, and much less to narrative reviews like ours. However, at the end of the introduction of the revised text, we provided some details about our literature search strategy.
Reviewer 3 Report
Very interesting topic , good presentation of pathogenic roles of alarmins in asthma
- The anti-alarmins therapy looks really promising, providing that real life data will confirm the good results from the clinical studies and that corresponding biomarkers will be identified
- Line 255 : the word “intense “ should be replaced and the described therapy does not correspond to maximal therapy, according to GINA step 4/5 ?
- Line 324 : the word “difuse “ has to be replaced with repeated or constant or chronic
- Line 319 : results of the SOURCE are missing, just the goal is mentioned
- Line 337 : Results of DESTINATION study regarding the impact on AAER are missing
- Some additional data regarding comparison with other biologicals already in clinical use might be useful
Author Response
We would like to thank very much the Reviewers for having carefully reviewed our manuscript, thus suggesting those changes which have significantly improved its overall quality. We have prepared a revised version of this paper, taking into account the comments of the 3 reviewers. We also include a point by point response to the criticisms raised by the Referees. The changes have been highlighted in yellow, and deletions have been evidenced in red. Therefore, we hope that our revised manuscript is now suitable for publication in Biomedicines.
REVIEWER 3
Line 255: the word “intense” has been replaced by the term “notable” in the revised manuscript.
Line 324: the word “diffuse” has been replaced by the term “chronic” in the revised manuscript.
Line 319: to our knowledge, the results of the SOURCE study have not yet been published.
Line 319: to our knowledge, the results of the DESTINATION study have not yet been published.
Unfortunately, scientific literature lacks head-to-head comparisons between different anti-asthma biologics.
Round 2
Reviewer 2 Report
A scenario review is out of date for a journal, which was dependent on author's preferences but not scientific.
Author Response
We would like to thank very much this Reviewer for his/her useful comments.